# Infection with *Clonorchis sinensis* (Cobbold, 1875) Metacercariae in Fish from the East Lake of Wuhan: Freshwater Fish in Urban Lakes May Act as Infection Sources of Liver Fluke

**DOI:** 10.3390/microorganisms12050898

**Published:** 2024-04-30

**Authors:** Jia-Nan Jiang, Hui-Fen Dong, Hou-Da Cheng, Hong Zou, Ming Li, Wen-Xiang Li, Gui-Tang Wang

**Affiliations:** 1Key Laboratory of Breeding Biotechnology and Sustainable Aquaculture (CAS), Institute of Hydrobiology, Chinese Academy of Sciences, Wuhan 430072, China; jjn18236960785@163.com (J.-N.J.); chenghouda@ihb.ac.cn (H.-D.C.); zouhong@ihb.ac.cn (H.Z.); liming@ihb.ac.cn (M.L.); liwx@ihb.ac.cn (W.-X.L.); 2University of Chinese Academy of Sciences, Beijing 100049, China; 3School of Medicine, Wuhan University, Wuhan 430072, China; hfdong@whu.edu.cn

**Keywords:** *Clonorchis sinensis*, the oriental liver fluke, fish-borne zoonotic trematodes, infection source, non-endemic area

## Abstract

The liver fluke disease caused by *Clonorchis sinensis* is one of the most serious food-borne parasitic diseases in China. Many freshwater fish and shrimps can be infected with *C. sinensis* metacercariae as the second intermediate hosts in endemic regions. Owing to the lack of infected humans and the good administration of pet dogs and cats in cities of non-endemic regions, few fish are expected to be infected with *C. sinensis* metacercariae in urban lakes. To determine the infection of *C. sinensis* metacercariae in freshwater fish and shrimps in urban lakes, a total of 18 fish species and one shrimp species were investigated in the East Lake of Wuhan City. Metacercariae were isolated by artificial digestive juice and identified using morphology and rDNA-ITS2 sequences. Five species of fish, *Pseudorasbora parva*, *Ctenogobius giurinus*, *Squalidus argentatus*, *Hemiculter leuciclus*, and *Rhodeus* spp., were infected with *C. sinensis* metacercariae. The overall prevalence of *C. sinensis* was 32.5%. The highest prevalence was found in *P. parva* with 57.9%, while *S. argentatus* exhibited the highest mean abundance (13.9). Apart from the *C. sinensis* metacercariae, four species of other trematode metacercariae were also identified across twelve fish species in total. Owing to the consumption of undercooked fish and feeding cats with small fish caught by anglers, there is a potential risk that the small fish infected with *C. sinensis* metacercariae may act as an infection source to spread liver fluke. Given the complete life cycle of *C. sinensis*, stray cats and rats were inferred to act as the important final hosts of *C. sinensis* in urban lakes in non-endemic areas.

## 1. Introduction

*Clonorchis sinensis* (Cobbold, 1875), also known as the oriental liver fluke, is one of the most important fish-borne zoonotic trematodes (FZT). Based on a national survey conducted in 2003 in China, 15 million people were estimated to be infected with *C. sinensis* in East Asia, of whom more than 12 million were in China [1], and the remainder were distributed across South Korea, Vietnam, and Russia [1,2,3]. In China, the major endemic areas were concentrated in four provinces: Guangxi and Guangdong in the south and Heilongjiang and Jilin in the northeast [4]. The consumption of raw or undercooked freshwater fish was considered to be responsible for the high prevalence in the endemic regions of East Asia [5].

There are three hosts in the life cycle of *C. sinensis*: freshwater snails as the first intermediate host, freshwater fish and occasionally shrimps as the second intermediate host, and human or carnivorous mammals (cats and dogs) as the definitive host. In the endemic regions, a high prevalence of *C. sinensis* metacercariae was commonly reported in some fish species in fishponds, lakes, and rivers [6]. The overall infection rate of *C. sinensis* metacercariae was 37.1%, and a higher than 50% prevalence was found in *Pseudorasbora parva* and *Ctenopharyngodon idelluse* in Guangdong Province [7], as well as a high prevalence in *P. parva* (45.3%) and in *Misgurnus anguillicaudatus* (41.2%) in Guangxi Province [8]. In Heilongjiang Province, a 19.9% overall prevalence was found in 3221 examined fish, and *P. parva* again exhibited the highest prevalence of 42.6% [9]. The habit of building toilets and pigsties next to fishponds in endemic areas contributed to the eggs ending up in pond water, which then caused high infection rates of *C. sinensis* metacercariae in freshwater fish [5].

In non-endemic regions, *C. sinensis* metacercariae were also detected in fish in fishponds, lakes, and rivers [10,11]. Domestic dogs and cats, usually fed raw fish, were considered the key final hosts to discharge feces with eggs of *C. sinensis* into waters [5,12,13]. Nowadays, in non-endemic regions, few people are infected with *C. sinensis*, and it is prohibited to discharge feces into urban lakes. Dogs and cats kept as pets are also well-administered from a veterinary standpoint. Thus, the prevalence of *C. sinensis* metacercariae in fish from urban lakes is expected to be low.

East Lake of Wuhan is one of the biggest urban lakes in China, covering an area of 33 square kilometers. The lake is isolated from the Yangtze River and other rivers [14], and more than 39 fish species were found in the lake [15]. The discharge of industrial wastewater and domestic sewage into the East Lake is not allowed. In early studies, *C. sinensis* metacercariae were found in fish collected from the fish market in Wuhan [16,17]. During the third survey of parasitic diseases in Hubei Province, conducted from 2014 to 2015, only one case of infection with *C. sinensis* was detected, in Chongyang City, far from Wuhan City [18]. In recent decades, pet cats and dogs are also well-administered around the East Lake. On the basis of these circumstances, we hypothesized that it is unlikely that *C. sinensis* metacercariae can be detected in freshwater fish and shrimps in the East Lake of Wuhan. To test this hypothesis, the infection of *C. sinensis* metacercariae was investigated in fish and shrimps collected from the East Lake.

## 2. Materials and Methods

### 2.1. Collection of Fish and Shrimps

Fish and shrimps were collected in multiple batches using a net or fish cage with a fine mesh in East Lake (113°41′–115°05′ E, 29°58′–31°22′ N) from April to July in both 2021 and 2022. The fresh fish and shrimps were kept at 5 °C and examined within two days.

### 2.2. Examination of Encysted Metacercariae

First, species of fish and shrimps were identified, and the total length and weight of each fish were measured. The head, scales, viscera, and bones of each fish were removed, and the remaining fish meat was collected for further examination. The muscle tissue of each fish was minced and mixed with artificial gastric juice (67.4 mL hydrochloric acid, 50 g pepsin, diluted with distilled water to 10 L; ten times the volume of the fish meat). The mixture was incubated for 5–8 h at 37 °C, and the digested fish meat was then filtered with a 40-mesh copper sieve. The fluid was placed into a 500 mL beaker for 20 min, and the supernatant was then replaced with a normal saline solution. This procedure was repeated more than 3 times until the supernatant was clear. Finally, the supernatant was discarded, and the sediment was placed into a glass dish in batches [7]. The presence of encysted metacercariae was examined under a stereomicroscope. Metacercariae in each fish were counted and preserved in 90% alcohol for morphological and molecular identification.

### 2.3. Morpholgocial Identification of Metacercariae

Based on the characteristics of suckers and excretory bladders, the metacercariae of *C. sinensis* were differentiated from other trematode species [19]. Other metacercariae were also grouped according to these morphological characteristics. According to the number of collected metacercariae, 1 to 11 metacercariae from each group were chosen for further molecular identification.

### 2.4. Molecular Identification of Metacercariae

The collected metacercariae were extracted using a DNA extraction kit (TIANGEN, Beijing, China), and the internal transcribed spacer (ITS) was used for species identification. Given more ITS-2 sequences of *C. sinensis* in GenBank, the species-specific primers CS1, 5′-CGAGGGTCGGCTTATAAAC-3′, and CS2, 5′-GGAAAGTTAAGCACCGACC-3′, were chosen to amplify the ITS-2 sequences of *C. sinensis* [20]. PCR amplification was conducted in 25 µL volumes, containing 1 µL of DNA template with 20 ng, 5 µL of 5 × Taq flexi buffer (pH 8.5), 2 µL of MgCl_2_ (25 mM), 2 µL of dNTP Mixture (2.5 mM), 0.5 µL of each primer (10 pmol/µL), and 0.13 µL of go Taq DNA polymerase (5 U/µL). The cycling parameters were as follows: 3 min at 94 °C, then 35 cycles of denaturation at 94 °C for 1 min, annealing at 53 °C for 1 min, an extension at 72 °C for 1 min, and a final extension of 5 min at 72 °C [20].

For the other trematode species, the ITS sequences were amplified using the primers BDI, 5′-GTCGTAACAAGGTTTCCGTA-3′, and BDII, 5′-TATGCTTAAATTCAGCGGGT-3′ [9]. PCR amplification was conducted in 25 µL volumes, containing 1 µL of DNA template, 5 µL of 5×Taq flexi buffer (pH8.5), 2 µL of MgCl_2_ (25 mM), 2 µL of dNTP Mixture (2.5 mM), 0.5 µL of each primer (10 pmol/µL), and 0.13 µL of go Taq DNA polymerase (5 U/µL). The cycling parameters were as follows: initial denaturation at 95 °C for 2 min, then 35 cycles of denaturation at 95 °C for 1 min, annealing at 50 °C for 1 min, an extension at 72 °C for 1.2 min, and a final extension of 5 min at 72 °C [9]. Each amplicon was examined by agarose gel (1.5%) electrophoresis and ethidium bromide staining. The positive products were sent to the company (Sangon Biotech, Shanghai, China) for sequencing.

Species of *C. sinensis* and other trematode metacercariae were then identified by blasting against the nucleotide database of GenBank using the obtained sequences.

### 2.5. Statistical Analysis

Owing to the limited fish sample size in one month, all the fish specimens collected at different months and years were put together for analysis of infection. Prevalence (infection rate) was defined as the number of fish infected with metacercariae of *C. sinensis* or other trematodes divided by the number of fish examined ×100%. Mean abundance was calculated as the total number of metacercariae divided by the number of fish examined [21].

## 3. Results

### 3.1. Identification of Clonorchis sinensis Metacercariae

Metacercariae with two equal-sized suckers, O-shaped dark excretory bladders, and brownish pigment granules were identified as *C. sinensis* (Figure 1A). ITS2 sequences of the 11 sequenced specimens were identical and also exhibited 100% sequence identity with a sequence belonging to *C. sinensis* available in GenBank (MN128618; Table 1).

### 3.2. Identification of Metacercariae of other Trematode Species

Except for *Clonorchis sinensis*, four forms of other trematode metacercariae were identified based on the shape of metacercariae, wall thickness, and the outline of the entity inside the wall (Figure 1B–E). ITS sequences of the four forms of metacercariae exhibited less than 98.1% identity (Table 2). On the basis of sequences available in GenBank, we inferred that form 1, form 2 and form 3 may belong to different species in the family Heterophyidae (Opisthorchiida) (Table 1), whereas form 4 exhibited the highest similarity (94.7%) with *Holostephanus* sp. in the family Cyathocotylidae (Diplostomida).

### 3.3. Infection with Clonorchis sinensis and Other Trematode Metacercariae in Fish and Shrimps

Among the 815 fish (18 species) examined, 265 fish specimens belonging to five species were found to be infected with *C. sinensis*. A total of 1622 metacercariae were detected, and the overall prevalence was 32.5%. The highest prevalence (57.9%) was found in *P. parva*, followed by *Ctenogobius giurinus*, *Rhodeus* spp., *Squalidus argentatus*, and *Hemiculter leucisculus*. The highest mean abundance (13.9 ± 2.8) was found in *S. argentatus*. No metacercariae were detected in the 219 shrimps, all of which belonged to the same species *Macrobrachium nipponense* (Table 3).

Twelve species of fish were found to be infected with four other types of trematode metacercariae. Their total number was 672, and their prevalence ranged from 6.7% to 100% (Table 3). 

## 4. Discussion

In East Lake of Wuhan, five fish species were unexpectedly found to be infected with *C. sinensis* metacercariae: *P. parva*, *C. giurinus*, *Rhodeus* spp., *S. argentatus*, and *H. leucisculus*. As the second intermediate host, freshwater fish are usually infected with metacercariae of *C. sinensis* [5]. The highest prevalence (35.1%) of *C. sinensis* metacercariae in fish was found in China, followed by Korea (29.7%) and Vietnam (8.4%) [22]. In China, more than 100 species of fish (59 genera in 15 families) and 4 species of shrimp are recognized as hosts [23]. Among these fish species, multiple small fish species, such as *P. parva*, *Abbottina sinensis*, *Saurogobio dabryi*, *Parapelecus argenteus*, and *Gnathopogon timberbis*, and some larger fish, such as *Ctenopharyngodon idellus*, *Cyprinus carpio*, *Carassius auratus*, and *Parabramis pekinesis*, are considered to be important second intermediate hosts of *C. sinensis* [5,13].

As in previous investigations, the topmouth gudgeon *P. parva* was infected with *C. sinensis* metacercariae with a high prevalence and mean abundance. The preferred water layer inhabited by topmouth gudgeon, and its thin skin, were proposed as the primary factors that make this species highly susceptible to *C. sinensis* [9]. *Hemiculter leucisculus* and *Rhodeus* spp. were also previously recognized as common second intermediate hosts [8,9], but this is the first record of *C. giurinus* and *S. argentatus* as the second intermediate hosts of *C. sinensis*.

The consumption of raw or undercooked freshwater fish is considered to be the main transmission route of *C. sinensis* in human infections [1,5]. Although small fish are generally not used to make sashimi, small fish fried with flour are commonly consumed in China. If such fish is undercooked, it may be dangerous to eat it. In addition, East Lake is popular with recreational anglers, who often use the caught small fish to feed cats, so the cats may become infected with *C. sinensis*. It is also possible that humans may become infected via the accidental ingestion of *C. sinensis* metacercariae on their hands after handling infected fish [24], even if the probability of this event is considered rare. Therefore, there is a potential risk that infected fish in East Lake may act as infection sources, transmitting *C. sinensis* to the final hosts.

Furthermore, the topmouth gudgeon *P. parva* was accidentally introduced into Europe and Africa with commercial fish and became an invasive fish in the world [25]. Fish species in Rhodeinae were caught and usually bred as ornamental fish in China. Consequently, the introduction and trade of *P. parva* and *Rhodeus* spp. also inevitably spread *C. sinensis* metacercariae worldwide.

Besides *C. sinensis* metacercariae, other trematode metacercariae were frequently detected in freshwater fish. There were 16 species of zoonotic trematode metacercariae encysted in freshwater fish in Korea, and the FTZ belonged to seven families, such as Opisthorchiidae, Heterophyidae, Echinostomatidae, Clinostomidae, Cyathocotylidae, Cryptogonimidae, and Bucephalidae [19]. Metacercariae of *Metorchis orientalis* (Opisthorchiidae), *Haplorchis taichui*, *H. pumilio*, and *Centrocestus formosanus* (Heterophyidae) were found in *P. parva* and *Rhodeus* spp. in China [26]. In the present study, three species in Heterophyidae and one species in Cyathocotylidae were identified in twelve fish species using the ITS sequence marker. The high prevalence (6.7–100%) of other trematode metacercariae suggested that more attention should be paid to the infection of other FTZ in freshwater fish.

The finding of *C. sinensis* metacercariae in freshwater fish suggested that *C. sinensis* could complete its life cycle in East Lake, i.e., that the first intermediate hosts and the final hosts also populate this urban lake. In China, eight species of freshwater snails were recognized as the first intermediate host for *C. sinensis*: *Alocinma longicornis*, *Parafossarulus striatulus*, *P. sinensis* (Hydrobiidae), *Semisulcospira cancellata* (Melaniidae), *Bithynia fuchsianus*, *B. robustus* (Bithyniidae), *Melanoides tuberculata* (Thiaridae), and *Assiminea lutea* (Assimineidae) [5]. *Alocinma longicornis*, *P. striatulus*, *P. sinensis*, and *B. fuchsianus* were considered to be important hosts in Hubei Province [27]. During previous surveys of macrozoobenthos, *A. longicornis* and *P. striatulus* were found in East Lake [28]. The two snail species may act as the first intermediate hosts of *C. sinensis*.

In endemic regions, an important transmission route is the release of *C. sinensis* eggs into the external environment via the feces of infected humans or other definitive hosts. The prevalence of *C. sinensis* was high in cats and dogs in southern Chinese endemic regions [5]. In northern endemic regions, the prevalence was higher in cats than in dogs, and pigs were infected at lower levels [13]. In a recent survey of parasitic diseases in Hubei Province, only one case of human infection by *C. sinensis* was found in Chongyang City, far from Wuhan [18]. In addition, pet cats and dogs were well-administrated around East Lake, and the discharge of municipal wastewater including feces into the East Lake was forbidden. However, stray cats were occasionally seen around the lake. As there is no investigation of *C. sinensis* in other final hosts in Wuhan, we can only speculate that stray cats act as the definitive hosts to release eggs of *C. sinensis* into the East Lake of Wuhan.

Aside from cats, dogs, and pigs [12], rats (*Rattus norvegicus*) were also viable definitive hosts of *C. sinensis* under experimental conditions [7,13,29]. Indeed, during the epidemiological investigation of *C. sinensis* in Hubei Province, *C. sinensis* eggs were detected in the feces of rats [27]. The Asian house rat (*Rattus tanezumi*) is widely distributed in China and it can be found along river banks [30]. It is omnivorous, with meat and fish also included in its diet [31]. We frequently observed the feces of rats along the banks of East Lake. On this basis, we hypothesize that rats may also act as a reservoir for *C. sinensis* in urban lakes in non-endemic areas.

## 5. Conclusions

*Clonorchis sinensis* metacercariae were detected in five fish species from East Lake of Wuhan. Given the consumption of undercooked fried fish and feeding cats with small fish caught by anglers, there is a potential risk of infection for humans. The high dispersal ability of *P. parva* and *Rhodeus* spp. bred as ornamental fish may also contribute to the spread of *C. sinensis* metacercariae around the world. In addition, the finding of *C. sinensis* metacercariae suggested that this liver fluke can complete the life cycle in the East Lake of Wuhan. We hypothesize that stray cats and rats may act as the final or reservoir hosts of *C. sinensis* in urban lakes in non-endemic areas. To confirm this hypothesis, domestic cats and dogs, stray cats, and wild rats in this area should be investigated for infection with *C. sinensis* in the future. 

## Figures and Tables

**Figure 1 microorganisms-12-00898-f001:**
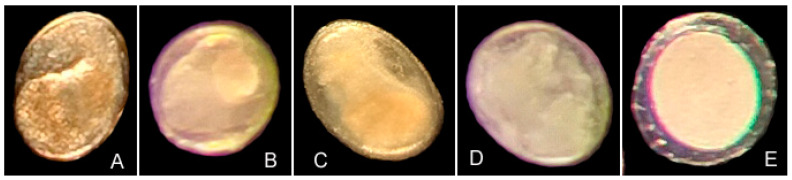
Morphological characteristics of *Clonorchis sinensis* (**A**) and four forms of other trematode metacercariae ((**B**), Form 1; (**C**), Form 2; (**D**), Form 3; (**E**), Form 4) collected from freshwater fish in the East Lake of Wuhan, China.

**Table 1 microorganisms-12-00898-t001:** Newly sequenced ITS2/ITS sequences of *Clonorchis sinensis* and four forms of other trematode metacercariae and their top similarity hits in GenBank.

Metacercariae Type	*C. sinensis*	Form 1	Form 2	Form 3	Form 4
Accession numbers	PP060703-PP060713	PP060714-PP060716	PP060717	PP060719-PP060720	PP060721
Top hits					
Similarity	100%	84.8%	83.6%	84.9–86.6%	94.7%
Species	*C. sinensis*	*Cryptocotyle lingua*	*Cryptocotyle lingua*	*Euryhelmis costaricensis*	*Holostephanus* sp.
Family	Opisthorchiidae	Heterophyidae	Heterophyidae	Heterophyidae	Cyathocotylidae
Accession number in GenBank	MN128618.1	MZ595806.1	MZ595806.1	AB521800.1	MT668948.1

**Table 2 microorganisms-12-00898-t002:** Pairwise identity values (%) for ITS rDNA sequences of the four forms of other trematode metacercariae.

Sample	Form 1-1	Form 1-2	Form 1-3	Form 2	Form 3-1	Form 3-2
Form 1-1						
Form 1-2	100					
Form 1-3	100	100				
Form 2	96.99	96.99	96.99			
Form 3-1	98.02	98.02	98.02	95.48		
Form 3-2	98.11	98.11	98.11	95.59	99.91	
Form 4	85.37	85.37	85.37	82.57	85.37	85.37

**Table 3 microorganisms-12-00898-t003:** Infection of *Clonorchis sinensis* and other trematode metacercariae in freshwater fish and shrimp from East Lake of Wuhan, China. N, the number of examined fish; ML, the mean fish body length; P, the prevalence of *C. sinensis* metacercariae; MA, the mean abundance of *C. sinensis* metacercariae per fish or in 5 g of fish meat; PT, the prevalence of metacercariae of other trematodes.

Fish Species	N	ML (cm)	P (%)	MA	PT (%)
*Cyprinus carpio*	29	7.5	0		20.7
*Carassius auratus*	52	10.8	0		7.7
*Hypophthalmichthys molitrix*	12	33.2	0		0
*Aristichthys nobilis*	15	15.9	0		25.0
*Ctenopharyngodon idella*	15	20.9	0		0
*Pseudorasbora parva*	300	4.3	57.9	3.5 ± 0.8	6.7
*Rhodeus* spp.	120	5.5	45.0	13.7 ± 2.5	33.6
*Squalidus argentatus*	56	8.2	28.6	13.9 ± 2.8	25.0
*Hemiculter leucisculus*	73	9.4	19.2	1.8 ± 0.3	25.0
*Ctenogobius giurinus*	15	4.1	46.7	3.1 ± 0.4	13.3
*Odontobutis obscurus*	12	3.5	0		57.1
*Misgurnus anguillicaudatus*	39	9.8	0		0
*Paramisgurnus dabryanus*	12	13.7	0		0
*Channa argus*	13	29.8	0		0
*Culter alburnus*	14	16.3	0		21.4
*Paracanthobrama guichenoti*	12	19.3	0		0
*Tachysurus fulvidraco*	15	4.8	0		100
*Gambusia affinis*	11	3.1	0		37.5
*Macrobrachium nipponense*	219	-	0		0
Total	1034			1622	672

## Data Availability

Data are contained within the article.

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
