# Peer review of "Infection with Clonorchis sinensis (Cobbold, 1875) Metacercariae in Fish from the East Lake of Wuhan: Freshwater Fish in Urban Lakes May Act as Infection Sources of Liver Fluke"

_microorganisms, 2024, doi:10.3390/microorganisms12050898_

Round 1
Reviewer 1 Report
Comments and Suggestions for Authors
Abstract: write the solid conclusion of the findings.
Introduction: the content in the introduction is well written.
Materials and methods:
Lines 77-79, add environmental parameters such as water parameters, water temperature, pH etc.
Lines 82-85, Please add pictures of fish and shrimps showing clinical signs of the parasite.
Lines 92-93, how were metacercariae and tissues preserved for DNA extraction? What kind of solution was used to storage the samples.
Lines 105-106, add concentration of DNA in 1 μl of DNA template. What was the size of the amplified PCR product?
Lines 121-122, sequences of metacercariae must be deposited to the NCBI databank or other seq repository and the accession numbers of the submitted sequences must be provided in the manuscript.
Statistical analysis: Please add more details about the statistical analysis.
Figure 1: Add scale bar for each metacercariae picture.
Ethical statement: The authors used fish and shrimps for the sampling but ethical approval number is not provided. This is a serious concern for this study.
Author Response
Abstract: write the solid conclusion of the findings.
Response: Yes, the conclusion is based on the findings.
Introduction: the content in the introduction is well written.
Response: Thanks.
Materials and methods:
Lines 77-79, add environmental parameters such as water parameters, water temperature, pH etc.
Response: Owing to the fish and shrimps were collected in multiple batches in different months, environmental parameters are different and unstable. So they were not provided here.
Lines 82-85, Please add pictures of fish and shrimps showing clinical signs of the parasite.
Response: Fish infected trematode metacercariae in muscle usually did not show clinical signs.
Lines 92-93, how were metacercariae and tissues preserved for DNA extraction? What kind of solution was used to storage the samples.
Response: Thank you for the reminder. It was revised as “Metacercariae in each fish were counted and preserved in 90% alcohol for morphological and molecular identification.”
Lines 105-106, add concentration of DNA in 1 μl of DNA template. What was the size of the amplified PCR product?
Response: Thanks. The concentration of DNA template was added. The ITS-2 sequence of C. sinensis is about 300 bp in length, and the ITS sequence of other trematode is about 1000 bp in length.
Lines 121-122, sequences of metacercariae must be deposited to the NCBI databank or other seq repository and the accession numbers of the submitted sequences must be provided in the manuscript.
Response: Yes, they were presented in Table 1.
Statistical analysis: Please add more details about the statistical analysis.
Response: Only prevalence and mean abundance were calculated in this study. Significant differences in prevalence and abundance were not performed between different fish species. Perhaps the statistical analysis is unnecessary to compare significant differences between fish species.
Figure 1: Add scale bar for each metacercariae picture.
Response: Sorry, due to our negligence, the metacercariae were photographed without scale bar.
Ethical statement: The authors used fish and shrimps for the sampling but ethical approval number is not provided. This is a serious concern for this study.
Response: Thanks for the reminder. Added.
Reviewer 2 Report
Comments and Suggestions for Authors
Overall recommendation
Seasonality of sampling and the possible influence of this parameter on general infection rate and specific of the different positive fish species was not considered at all. Please consider rewriting some parts of the material and methods, results and discussion section considering this aspect, which contextually was not reported as one limitation of the study in the conclusion section. This aspect should be considered in this kind of study in each case, if or if not influences the infection incidence.
Title
Please add the eponym to the investigated species name (also valid for the first mention in the main text.
Keywords
Try to repeat words already reported in the title substituting them with other related ones.
Material and Methods
Please add more sampling details to paragraph 2.1 (es, sampling per month, the total number of samples per season, etc)
Lines 98-99: "1 to 11 metacercariae from each group were chosen for further molecular identification" depending on what? Please better argue this sentence explaining this variable number of analysed metacercariae per group.
I am in trouble with the procedure exposed in paragraph 2.5, indeed the authors consider as a sample 5g of muscle but previously reported that they collected this quantity just for fish larger than 50g, while for the smaller, the total amount collected muscle was not considered, based on this section. This could lead to a bias when considering all the samples as equal. Please take care to expose better and clarify this aspect. I suggest considering the samples as they are, without normalization to 5g which appears not uniformly applied to all samples.
Discussion
Lines 189-191: I suggest adding ", even if the probability of this event is considered rare" at the end of this sentence.
Lines 226-228: I understand that stating about possible wrong management of wastewater, domestic animals, and humans could also be a speculation, but it should be considered as a pair of the stray animals cause. While rats are, in my opinion, a possible primary source of infection considering their life traits, please state that all these possible causes need further specific studies to well-assess the cause of the abnormal incidence of infection found (as stated in the conclusion section as next scenarios).
Best regards
Author Response
Reviewer 2:
Seasonality of sampling and the possible influence of this parameter on general infection rate and specific of the different positive fish species was not considered at all. Please consider rewriting some parts of the material and methods, results and discussion section considering this aspect, which contextually was not reported as one limitation of the study in the conclusion section. This aspect should be considered in this kind of study in each case, if or if not influences the infection incidence.
Response: Thank you for the good suggestion. Yes, sampling seasonality is an important factor to influence the infection parameter, such as prevalence and abundance. Owing to difficulties in fish sampling at the urban lake, species and number of collected fish are not enough in one month. Thus all the fish samples obtained were put together for analysis. In addition, the metacercariae were usually found in fish flesh, and they were less affected by seasonality. As your suggestions, sampling date and treatment of fish sample at different months were added in Materials and Methods part.
It was revised as followings: In collection of fish and shrimps, fish and shrimps were collected in multiple batches using net or fish cage with a fine mesh in East Lake from April to July both in 2021 and 2022. In statistical analysis, Owing to limited fish sample size in one month, all the fish specimens collected at different months and years were put together for analysis of infection.
Title
Please add the eponym to the investigated species name (also valid for the first mention in the main text.
Response: Thanks for reminder. The eponym followed the species name was added to the title and the main text, such as Clonorchis sinensis (Cobbold, 1875).
Keywords
Try to repeat words already reported in the title substituting them with other related ones.
Response: It was changed as “Clonorchis sinensis, the oriental liver fluke, fish-borne zoonotic trematodes, infection source, non-endemic area”.
Material and Methods
Please add more sampling details to paragraph 2.1 (es, sampling per month, the total number of samples per season, etc)
Response: Sampling details were added as the above.
Lines 98-99: "1 to 11 metacercariae from each group were chosen for further molecular identification" depending on what? Please better argue this sentence explaining this variable number of analysed metacercariae per group.
Response: According to number of the collected metacercariae, 1 to 11 individuals from each group were chosen for further molecular identification.
I am in trouble with the procedure exposed in paragraph 2.5, indeed the authors consider as a sample 5g of muscle but previously reported that they collected this quantity just for fish larger than 50g, while for the smaller, the total amount collected muscle was not considered, based on this section. This could lead to a bias when considering all the samples as equal. Please take care to expose better and clarify this aspect. I suggest considering the samples as they are, without normalization to 5g which appears not uniformly applied to all samples.
Response: Actually, only few fish is greater than 50g. So it is revised as “The head, scales, viscera and bones of each fish were removed, and the remaining fish meat was collected for further examination”.
Discussion
Lines 189-191: I suggest adding ", even if the probability of this event is considered rare" at the end of this sentence.
Response: Thanks. Done.
Lines 226-228: I understand that stating about possible wrong management of wastewater, domestic animals, and humans could also be a speculation, but it should be considered as a pair of the stray animals cause. While rats are, in my opinion, a possible primary source of infection considering their life traits, please state that all these possible causes need further specific studies to well-assess the cause of the abnormal incidence of infection found (as stated in the conclusion section as next scenarios).
Response: Thank you for the valuable suggestions. It was revised as “To confirm this hypothesis, domestic cats and dogs, stray cats and wild rats in this area should be investigated for infection with C. sinensis in the future.” in the Conclusions part.
Reviewer 3 Report
Comments and Suggestions for Authors
1) The present research project belongs to local public health issues with liver fluke, so it will be value for human health. However, the text is too long, especially, the part "Introduction".
2) The title and its legend is sited upper side of the photographs of the fig. 1. According to the common rule of scientific paper, these should be shifted to the bottom with scales (bars and mm etc.).
3) But that is not fatal. The authors have to show each morphological characteristics for the types (?) A to E; for example suckers and guts etc..
4) And, they must show the relationship between the types (?) in Fig. 1 and the forms 1 to 4 in the tables 1 and 2, respectively.
Author Response
1) The present research project belongs to local public health issues with liver fluke, so it will be value for human health. However, the text is too long, especially, the part "Introduction".
Response: Detailed infection rate in some fish was removed in the 2nd paragraph of the Introduction part.
2) The title and its legend is sited upper side of the photographs of the fig. 1. According to the common rule of scientific paper, these should be shifted to the bottom with scales (bars and mm etc.).
Response: Thank you for the reminder. Revised.
3) But that is not fatal. The authors have to show each morphological characteristics for the types (?) A to E; for example suckers and guts etc.
Response: Yes, that is a good suggestion. But it is difficult to give each form with accurate name.
4) And, they must show the relationship between the types (?) in Fig. 1 and the forms 1 to 4 in the tables 1 and 2, respectively.
Response: Thanks. It was revised in the legend as the suggestion.
Round 2
Reviewer 1 Report
Comments and Suggestions for Authors
The authors have made all the necessary edits. The manuscript now appears much better readable.
Author Response
Thank you for your valuable suggestions.
Reviewer 2 Report
Comments and Suggestions for Authors
Dear Authors,
thank you for seriously considering my previous comments in revising your manuscript. I have just one comment about seasonality. In my opinion, even if your data are not uniform month by month, at least grouping among cold and warm seasons should be provided to enhance the value of your results, even if they do not show significant differences the readers should evaluate them by themselves through showed data.
Best regards
Author Response
Yes, you are right. Usually temperature has great effect on development, transmission and infection of parasites. In this study, the fish was collected from April to July, which is spring and early summer. It is also the warm season. In addition, the fish sample size is small in some months, such as in April and July. So it is difficult to group the fish sample into warm and cold season.
Thank you for your valuable suggestions!
Reviewer 3 Report
Comments and Suggestions for Authors
OK asa
Author Response
Thank you for your valuable suggestions to improve the manuscript.